# OmniEval: A Benchmark for Evaluating Omni-modal Models with Visual, Auditory, and Textual Inputs

## Abstract

In this paper, we introduce OmniEval, a benchmark for evaluating omni-modality models like Qwen2.5-Omni and MiniCPM-O 2.6, which encompasses visual, auditory, and textual inputs. Compared with existing benchmarks, our OmniEval has several distinctive features: (i) Full-modal collaboration: We design evaluation tasks that highlight the strong coupling between audio and video, requiring models to effectively leverage the collaborative perception of all modalities; (ii) Diversity of videos: OmniEval includes 780 audio-visual synchronized videos, 255 Chinese videos and 525 English videos; (iii) Diversity and granularity of tasks: OmniEval contains 2411 question-answer pairs, comprising 1278 open-ended questions and 1133 multiple-choice questions. These questions are divided into 3 major task types and 12 sub-task types to achieve comprehensive evaluation. Notably, we introduce a refined video localization task (*i.e.*, Grounding) designed to test precise spatio-temporal understanding. We evaluate several representative omni-modal models on OmniEval to demonstrate its utility. We hope that our OmniEval can provide a platform for evaluating the ability to construct and understand coherence from the context of all modalities.

## 1 Introduction

The pursuit of Artificial Intelligence (AI) systems capable of emulating human-like understanding of the world has catalyzed significant advancements in models that process information from multiple modalities (Radford et al., 2021; Alayrac et al., 2022; Li et al., 2023a). These Multimodal Large Language Models (MLLMs) have demonstrated remarkable potential in tasks like image captioning, visual question answering, and text-to-image generation (Team, 2024; 2025b). However, a prevailing trend is the development of "omni-modal models" capable of concurrently processing and understanding information from all three modalities: visual, auditory, and textual (Xu et al., 2025; Fu et al., 2025; Cheng et al., 2024; OpenBMB Team, 2025). Such models aim to more comprehensively simulate human perception and cognition of the world, laying the foundation for more complex and realistic application scenarios, including intelligent assistants, robotic interaction, and content creation.

Despite the promising application prospects of omni-modal models, comprehensively and effectively evaluating their integrated capabilities remains a critical unresolved issue. Existing multimodal benchmarks predominantly focus on combinations of one or two modalities (e.g., vision-text or audio-text) or fail to adequately reflect the deep coupling and synergistic effects among multimodal information in their task design (Li et al., 2025b; Hong et al., 2025b). For instance, some existing benchmarks may focus on static visual content paired with audio, thereby inadequately assessing the understanding of dynamic visual events crucial for real-world scenarios (Li et al., 2025b). Others, while offering a broader range of tasks, might be limited to a single language, thus failing to evaluate a model's multilingual capabilities (Hong et al., 2025b). Consequently, these benchmarks often fall short in evaluating the deep, synergistic understanding that arises from the concurrent integration of dynamic visual, auditory, and textual cues across diverse linguistic contexts. They may also lack the task diversity or the fine-grained evaluation mechanisms, such as precise temporal grounding, necessary to truly probe how omni-modal models interpret and fuse these distinct information streams to achieve a holistic understanding. Particularly for questions requiring models to

simultaneously integrate visual dynamics, sound events, and associated text (such as subtitles or dialogue) for accurate answers, current evaluation methods often prove inadequate. Moreover, existing models still face substantial challenges in real-world understanding, which further underscores the necessity of constructing a more comprehensive and challenging evaluation benchmark.

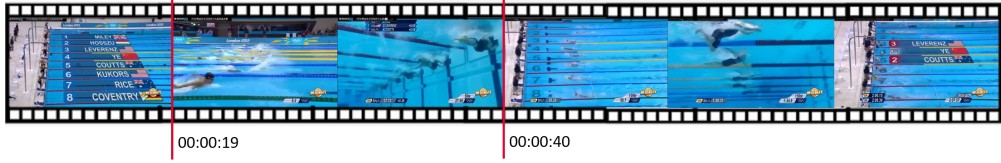

**Question**: 视频中对话出现开始之后就已经拉开差距了这句话，到选手们游至泳池对面期间时间段是多少？ 请给出视频中的具体开始时间和结束时间。(In the video, when the commentary states 'the gap had already widened right from the start', what is the exact time duration between this point and when the swimmers reach the opposite end of the pool? Please specify the precise start and end timestamps.)
**Answer**: 00:00:19 - 00:00:40

Figure 1: A grounding example in OmniEval. OmniEval requires integrating both visual and auditory signals to provide accurate answers for certain questions, while also incorporating fine-grained understanding tasks such as grounding.

To address this critical evaluation gap, we introduce OmniEval, a novel benchmark specifically designed to rigorously evaluate omni-modal models that jointly process and reason across visual, auditory, and textual inputs, supporting both Chinese and English languages. OmniEval possesses several distinctive features: 1) **Full-modal Collaborative Evaluation:** We have meticulously designed evaluation tasks that emphasize the strong coupling between audio and video, requiring models to effectively leverage the collaborative perception of all modalities for correct answers (Figure 1). This transcends evaluation approaches that merely sum individual unimodal understanding capabilities. 2) **Diverse Videos and Task Scenarios:** OmniEval comprises 780 audio-visual synchronized video clips, including 255 Chinese videos and 525 English videos. These videos ensuring broad coverage of evaluation scenarios. 3) **Diverse and Fine-grained Task Design:** OmniEval contains 2411 question-answer pairs, consisting of 1278 open-ended questions and 1133 multiple-choice questions. These questions are divided into 3 major task types and 12 sub-task types, aiming for a comprehensive assessment of model capabilities. Notably, we introduce a more fine-grained video localization task, termed Grounding (Figure 1), to precisely evaluate the model's ability to locate information in the temporal dimension.

Based on OmniEval, we have conducted extensive evaluations of various state-of-the-art omni-modal models. The experimental results indicate that existing models face significant challenges in understanding real-world information. This clearly demonstrates the challenging nature of OmniEval and the urgent need to enhance the capabilities of current models.

The main contributions of this paper are as follows:

- We construct and release OmniEval, a novel and comprehensive omni-modal evaluation benchmark suite, that focuses on assessing models' synergistic understanding and processing of visual, auditory, and textual information, with bilingual support including Chinese and English.

- OmniEval introduces diverse video content and fine-grained task types, particularly establishing tasks that emphasize strong audio-visual coupling and precise temporal localization (*i.e.*, Grounding), offering a new perspective for a more comprehensive measurement of model capabilities.

- We conduct extensive testing and analysis of current mainstream omni-modal models on OmniEval, providing valuable baselines, revealing the deficiencies of existing models in real-world understanding, and offering insights for future research directions.

We hope that OmniEval will serve as an important benchmark to drive the development of omni-modal models, encouraging researchers to build more powerful models capable of understanding and constructing coherence from the context of all modalities. Our dataset and evaluation code are publicly available to foster further research in the community.

## 2  RELATED WORK

### 2.1  MULTIMODAL LARGE LANGUAGE MODELS

Recent advancements in large language models (LLMs) have demonstrated significant improvements across a wide range of natural language processing (NLP) tasks (Team, 2025a; 2024; 2020; 2025b). These models, characterized by their deep architectures and extensive pretraining on massive corpora, have consistently outperformed traditional methods in benchmarks such as question answering (Bonfigli et al., 2024), machine translation (Zhang & Shafiq, 2024), summarization (Bonfigli et al., 2024), and text generation (Team, 2024; 2020). There has been an increasing interest in incorporating multiple modalities into large language models (LLMs), with the goal of enhancing their capabilities beyond textual processing alone. (Li et al., 2023a; Liu et al., 2023; Xu et al., 2025; Fu et al., 2025) In the visual domain, raw images are processed through specialized visual encoders to obtain high-level features, while in the audio domain, raw waveforms are first sampled and then encoded using dedicated audio encoders. These modality-specific representations are subsequently aligned with textual tokens using intermediate modules such as Querying Transformers (Q-Former) (Li et al., 2023a), Multi-Layer Perceptrons (MLPs) (Liu et al., 2023), or other alignment techniques (Wang et al., 2024a). This semantic alignment enables the fusion of heterogeneous inputs into a unified representation space. Leveraging the generative capabilities of LLMs, the resulting multimodal architecture achieves strong performance across a range of tasks, including image captioning (Liu et al., 2023; Wang et al., 2024a), visual and spoken question answering, audio captioning (Chu et al., 2023), and multimodal dialogue (Fu et al., 2025). In addition, some models have attempted to integrate both visual and auditory understanding into a single, unified framework, thereby creating omni-modality models (Xu et al., 2025; Fu et al., 2025; Cheng et al., 2024; OpenBMB Team, 2025). However, evaluating the performance of such models presents a significant challenge, as it requires designing tasks that simultaneously involve multiple modalities. The lack of standardized evaluation metrics and benchmarks for these models remains an open problem, and addressing this issue is critical for advancing the development and comparison of multimodal AI systems.

Table 1: The comparison of various benchmarks encompasses several key aspects: modality involved (**Modality**), languages involved (**Language**), format of Q&A pair (**QA Format**), whether including event grounding task (**Grounding**), the source of videos (**Video Sources**), the method of generating questions and answers (**QA Generation**) and the number of Q&A pairs (**No. of QA Pairs**). A, V and I for modality represent audio, video and image, respectively. OE indicates open-ended questions, MC indicates multiple-choice questions.

| Feature | Modality | Language | QA Format | Grounding | Video Sources | QA Generation | No. of QA Pairs |
|---|---|---|---|---|---|---|---|
| OmniBench (Li et al., 2024) | I+A | EN | MC | No | No | Manual | 1143 |
| MMbench-Video (Fang et al., 2024) | V | EN | OE | No | YouTube | Manual | 1998 |
| DeVE-QA (Qin et al., 2024) | V | EN | Limited OE | Yes (Grounding required) | ActivityNet | LLM + Manual | 78000 |
| Video-MME (Fu et al., 2024) | V+A | EN | MC | No | YouTube | Manual | 2700 |
| WorldSense (Hong et al., 2025a) | V+A | EN | MC | Yes (Coarse-grained) | YouTube, MusicAVQA | Manual | 3172 |
| LongVALE (Geng et al., 2024) | V+A | EN | No QA | Yes | YouTube | LLM + Manual | 0 |
| StreamingBench (Lin et al., 2024) | V+A | EN | OE | No | YouTube | LLM + Manual | 4500 |
| CG-Bench (Chen et al., 2024) | V+A | EN | MC | No | YouTube, BiliBili | Manual Curation | 12129 |
| **OmniEval** | V+A | EN & CN | MC & OE | Yes (Fine-grained) | YouTube, Youku | LLM + Manual | 2411 |

### 2.2  MULTIMODAL BENCHMARKS

Recently, a wide range of benchmarks exist to evaluate the understanding and reasoning capabilities of large language models (Zellers et al., 2019; Wang et al., 2024b; Hendrycks et al., 2021; Cobbe et al., 2021). In the visual domain, prior works assess model performance across multiple dimensions, including object recognition (Young et al., 2014; Plummer et al., 2017; Li et al., 2023b) and localization (Kazemzadeh et al., 2014; Yu et al., 2016), image-based question answering (Goyal et al., 2017; Antol et al., 2015; Zhang et al., 2016; Liu et al., 2024; Gurari et al., 2018), and visual commonsense reasoning (Masry et al., 2022; Lu et al., 2024; Singh et al., 2019). Similarly, in the auditory domain, existing benchmarks focus on tasks such as automatic speech recognition (Hernandez et al., 2018; Conneau et al., 2022; Panayotov et al., 2015; Bu et al., 2017; Zhang et al., 2022; Chen et al., 2021), audio-based question answering (Joshi et al., 2017; Lipping et al., 2022; Nachmani et al., 2024; Yang et al., 2024), and audio scene understanding (Poria et al., 2019; Chen et al., 2018; Nagrani et al., 2017; Yang et al., 2024). These benchmarks serve as essential tools for measuring

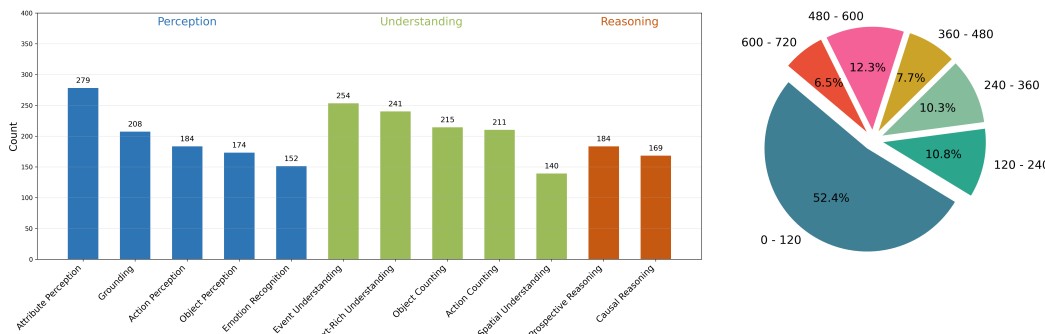

Figure 2: The left diagram depicts task question quantity grouped by functional categories in OmniEval. Blue bars represent perception-related tasks, green indicates information processing tasks, and orange denotes higher-order reasoning tasks. The right diagram depicts video duration distribution. The unit of the number is second. OmniEval covers videos of various lengths. The average video length in the dataset is 214 seconds.

the effectiveness of multimodal models in real-world applications, enabling systematic comparisons across different modalities and model architectures. For omni-modality models, the number of available benchmarks is limited, and most of them exhibit certain shortcomings. Some studies (Li et al., 2025b) provide static images along with speech to test models' abilities in speech understanding and static scene perception, yet they overlook the model's capacity to process dynamic visual information. Other work (Hong et al., 2025b) focuses on testing models' understanding of both audio and video by providing video and audio inputs, but these benchmarks often lack diversity in testing scenarios, tasks, and languages. As a result, there is a need for more comprehensive and standardized evaluation frameworks that can better assess the full range of capabilities in omni-modality models, including their ability to handle dynamic multimodal inputs across varied real-world conditions. To address these issues, we propose OmniEval, a comprehensive benchmark designed specifically for evaluating the full range of capabilities in omni-modality models.

## 3 OMNIEVAL

To foster more comprehensive evaluation for omni-modal MLLMs, we introduce a new benchmark dataset specifically designed with multilingual support and a balanced mix of question formats. This chapter details the systematic pipeline developed for its construction, emphasizing methodological rigor and quality control. Our pipeline integrates automated data processing using large models with essential manual curation, aiming to create a challenging and reliable resource for evaluating Omni models across diverse cognitive tasks, including fine-grained event understanding inspired by the need for temporal localization.

### 3.1 DATA COLLECTION AND PREPROCESSING PIPELINE

This phase focused on assembling a diverse video collection and extracting the necessary textual modalities (captions and speech transcripts) to serve as the foundation for Q&A generation.

For the first step, we initiated the process by aggregating video information from established video benchmarks such as FineVideo (Farre et al., 2024) and Youku-mplug (Xu et al., 2023) in compliance with the license regulations. This hybrid sourcing strategy aimed to ensure broad coverage of topics, styles, and real-world scenarios, moving beyond the confines of specific dataset domains. The goal was to create a varied collection challenging models on multiple fronts.

For the second step, we acquired corresponding captions and subtitles for each identified video. When available from source benchmarks, existing high-quality caption tracks were utilized. For other videos, captions were obtained with MLLMs like Qwen2.5-VL-72B (Xu et al., 2025) or generated using appropriate methods, ensuring a textual description accompanied each video.

To capture the linguistic content within the audio track, we employed the Volcano Engine large model for Automated Speech Recognition (ASR). Accurate ASR was performed for all videos, configured for the primary languages present (Chinese and English), yielding transcripts of spoken content.

Then for the third step, Videos containing little or no spoken content (identified via metrics like word count or speech duration) were excluded based on the ASR transcripts. Specifically, as with FineVideo (Farre et al., 2024), we calculate the word sensitivity for each video and exclude those with subdensity less than 0.5. This ensures that the remaining videos possess meaningful linguistic information in the audio modality, complementing the visual stream. This step is vital as our subsequent Q&A generation leverages both captions and transcripts (Section 3.3), aiming to probe deeper audio-visual understanding rather than purely visual recognition.

### 3.2 Q&A Pair Generation and Annotation Pipeline

Leveraging the curated videos and their associated text, we implemented a multi-stage pipeline for generating and categorizing QA pairs.

#### 3.2.1 Automated Q&A Generation

We employed large models for automated Q&A generation, capitalizing on their ability to process multimodal context and formulate relevant questions. The process involved three stages:

(i) Open-Ended (OE) Generation: Models were prompted with both video captions and audio subtitles to generate OE questions and corresponding answers. This approach provides rich context, combining descriptive text with spoken dialogue/narration. Generating OE questions first allows for capturing more complex and nuanced aspects of the video content without the initial constraint of predefined answer choices.

(ii) Multiple-Choice (MC) Derivation: Subsequently, the generated OE pairs were used as input for another large model task: converting the OE question into an MC format. This involved generating plausible distractors alongside the correct answer derived from the OE pair. Including MC questions facilitates standardized evaluation protocols common in the field.

(iii) Removing those overly simple samples: To ensure the complexity and robustness of the benchmark, we rigorously evaluated the Q&A pairs using multiple large models, and systematically removed questions that could be answered correctly by all models. This process helps to maintain a high level of challenge within the benchmark.

Specifically, we have meticulously crafted two distinct categories of Q&A pairs tailored for **Grounding**: moment-based and time span-based. Moment questions zero in on pinpointing the precise instant when fleeting events unfold within the video, exemplified by queries like, "At what exact moment does the girl in red commence her speech within the frame?" Conversely, time span questions delve into the broader temporal context, seeking to identify the specific duration during which a particular event transpires, such as, "Over which interval in the video does the girl in red engage in delivering her speech?". Each grounding Q&A pair is categorized into either moment-based or the time span-based category and is assessed using different methods accordingly.

#### 3.2.2 Q&A Classification

Each generated Q&A pair (both OE and MC) was automatically classified into one of 12 predefined categories reflecting different cognitive skills: Grounding, Object Counting, Action Counting, Prospective Reasoning, Text-Rich Understanding, Event Understanding, Attribute Perception, Action Perception, Spatial Understanding, Causal Reasoning, Object Perception, and Emotion Recognition. This fine-grained classification enables nuanced analysis of model strengths and weaknesses across different facets of multimodal understanding. The inclusion of a grounding category specifically targets the model's ability to link answers to specific temporal moments in the video.

#### 3.2.3 Manual Curation and Quality Assurance

Recognizing the potential limitations of fully automated generation, we involved meticulous manual review and revision of all Q&A pairs by human annotators to guarantee:

Table 2: Format Distribution of Q&A pairs in OmniEval.

| Question Format | Num. |
|---|---|
| Open-Ended (OE) | 1278 |
| Multiple-Choice (MC) | 1133 |
| Total | 2411 |

Table 3: Language distribution of videos and Q&A pairs in OmniEval.

| Language | Videos Num. | Q&A Pairs Num. |
|---|---|---|
| Chinese (CN) | 255 | 898 |
| English (EN) | 525 | 1513 |
| Total | 780 | 2411 |

- Clarity: Refining question and answer wording for unambiguity.

- Relevance and Grounding: Confirming questions are pertinent and answerable from the video, not just based on model biases.

- Accuracy: Ensuring answers are factually correct based on video content.

- Judgement: To determine the number of modalities of information required to answer a question correctly and refine the task type of questions.

- Distribution: Given that the Q&A pair directly generated by large language models are unevenly distributed in terms of capability items, such as Grounding, Action Counting, Object Counting, we asked five people to watch the videos and write corresponding question-answer pairs.

### 3.2.4 BENCHMARK STATISTICS

The construction pipeline yielded a benchmark with a significant number of Q&A pairs distributed across different formats, task types, and languages.

As shown in Tables 2 and 3, our benchmark features a well-balanced distribution of OE and MC question formats, accommodating diverse evaluation criteria. This design enables performance analysis on both OE and MC questions when an LLM is available for OE evaluation. Conversely, without an LLM for OE assistance, the benchmark still facilitates a thorough analysis of MC question performance alone.

As shown in Figure2, the question–answer pairs are classified into 12 distinct types, enabling a fine-grained analysis of model performance across various cognitive skills. Special attention is given to the inclusion of a grounding task (208 pairs), which addresses the need for models to precisely localize information in the temporal dimension.

A key characteristic of our benchmark is its bilingual nature, encompassing both Chinese and English videos and Q&A pairs. This facilitates research in multilingual MLLM capabilities.

### 3.3 COMPARISON WITH EXISTING BENCHMARKS

Our benchmark introduces several distinguishing features compared to existing video understanding benchmarks, aiming to provide a more comprehensive evaluation tool for omni models. Table 1 provides a comparative overview.

Key differentiators of our benchmark include:

- Bilingual Support: Unlike many prominent benchmarks that are predominantly English-based (e.g., WorldSense, LongVALE, StreamingBench), our benchmark incorporates a significant volume of both English and Chinese videos and Q&A pairs. This facilitates direct evaluation and development of omni models for these two major languages.

- Emphasis on Open-Ended Questions: Many existing benchmarks heavily rely on MCQs for evaluation (e.g. WorldSense, DeVE-QA). Our benchmark provides a substantial number of OE questions (1278 pairs), allowing for a more in-depth assessment of omni models' generative capabilities, their ability to formulate detailed explanations, and their performance in scenarios that mimic natural human interaction more closely than restricted choice formats.

- Integrated Event Grounding: While benchmarks like LongVALE and DeVE-QA emphasize temporal understanding and event localization, our benchmark uniquely includes grounding as one of its 12 Q&A categories. This enables targeted evaluation of a model's ability to connect answers

to specific video segments, demonstrating comprehension beyond mere pattern matching. Although WorldSense features coarse-grained "Temporal Localization" multiple-choice questions (e.g., event at beginning/middle/end), our grounding questions offer both multiple-choice and open-ended formats, targeting exact video moments with greater granularity and an adaptive evaluation strategy.

By addressing these aspects, our benchmark aims to complement existing resources and provide a more nuanced and comprehensive platform for advancing MLLM research in video understanding.

Table 4: Overall performance on OmniEval. MNT indicates max new tokens. OE indicates open-ended QAs, MC indicates multiple-choice QAs.

| Methods | Params | Frames | MNT | Perception | | Understanding | | Reasoning | | Avg | | Overall |
|---|---|---|---|---|---|---|---|---|---|---|---|---|
| | | | | OE | MC | OE | MC | OE | MC | OE | MC | |
| Qwen2.5-Omni-7B (Xu et al., 2025) | 7B | 1fps | 1024 | 43.48 | 71.40 | 48.70 | 66.20 | 65.66 | 88.90 | 48.85 | 71.67 | 59.57 |
| Qwen2.5-Omni-3B (Xu et al., 2025) | 3B | 1fps | 1024 | 37.80 | 68.30 | 42.09 | 58.50 | 60.55 | 88.30 | 42.85 | 66.81 | 54.11 |
| Baichuan-Omni-1.5 (Li et al., 2025a) | 7B | 64 | 1024 | 31.58 | 66.20 | 35.14 | 61.20 | 48.74 | 85.40 | 35.53 | 66.81 | 50.23 |
| MiniCPM-O 2.6 (OpenBMB Team, 2025) | 8B | 64 | 1024 | 18.20 | 28.80 | 26.87 | 34.20 | 20.33 | 25.10 | 22.16 | 30.71 | 26.18 |
| VITA-1.5 (Fu et al., 2025) | 8B | 64 | 1024 | 5.72 | 12.93 | 9.72 | 7.49 | 4.29 | 8.77 | 7.20 | 9.80 | 8.42 |
| gemini-2.5-pro-preview-05-06 (Google & DeepMind, 2025) | - | 1fps | - | 56.42 | 69.40 | 63.95 | 68.30 | 81.32 | 60.20 | 63.15 | 67.52 | 65.20 |

# 4 EXPERIMENTS AND FINDINGS

In this section, we conduct a comprehensive evaluation of existing open-source multimodal MLLMs and Gemini 2.5 (Google & DeepMind, 2025) based on the proposed OmniEval benchmark. We begin by outlining the experimental setup and evaluation methodology, detailing the tasks, metrics and data used in our analysis. We then present an in-depth examination of the quantitative results, highlighting the strengths and weaknesses of different models across various modalities and tasks. Furthermore, we investigate several key factors that influence model performance, offering insights into the challenges and opportunities in multimodal understanding.

## 4.1 SETTINGS

To comprehensively evaluate the multimodal understanding capabilities of current models, we assess 6 fully multimodal models that integrate visual, textual, and auditory information. The evaluation configuration parameters are shown in Table 4.

For evaluation, we adopt different strategies for MC and OE Q&A pairs. For MC Q&A pairs, we directly determine whether the option output by the model is consistent with the ground truth. For OE questions, we leverage a powerful proprietary language model to assist in assessment. Specifically, we categorize Q&A pairs into grounding, counting and other tasks and utilize different assessment methods for different categories.

### 4.1.1 EVALUATION FOR GROUNDING OE Q&AS

For grounding open-ended tasks, we first leverage LLMs to extract temporal information from the model's output. Subsequently, we employ distinct strategies to evaluate various data types.

Specifically, for moment-based Q&A pairs, we've developed an adaptive evaluation method based on video frame extraction. When the number of extracted frames is low, the time intervals between adjacent frames become significantly larger. In such scenarios, precise alignment between

Table 5: Performance of the model on different language dimensions on OmniEval.

| Methods | Params | Frames | MNT | English | | | Chinese | | |
|---|---|---|---|---|---|---|---|---|---|
| | | | | OE | MC | ALL | OE | MC | ALL |
| Qwen2.5-Omni-7B (Xu et al., 2025) | 7B | 1fps | 1024 | 44.54 | 70.88 | 58.05 | 54.70 | 73.39 | 62.13 |
| Qwen2.5-Omni-3B (Xu et al., 2025) | 3B | 1fps | 1024 | 40.21 | 65.98 | 53.43 | 46.44 | 68.63 | 55.26 |
| Baichuan-Omni-1.5 (Li et al., 2025a) | 7B | 64 | 1024 | 36.97 | 64.43 | 51.06 | 33.57 | 71.99 | 48.84 |
| MiniCPM-O 2.6 (OpenBMB Team, 2025) | 8B | 64 | 1024 | 7.91 | 14.95 | 11.52 | 41.55 | 64.99 | 50.87 |
| VITA-1.5 (Fu et al., 2025) | 8B | 64 | 1024 | 2.22 | 0.39 | 1.28 | 13.99 | 30.35 | 20.50 |
| gemini-2.5-pro-preview-05-06 (Google & DeepMind, 2025) | - | 1fps | - | 61.30 | 68.56 | 65.02 | 65.66 | 65.27 | 65.50 |

the model's prediction and the true value may not be achievable. Therefore, we use a larger threshold to evaluate the model's output, allowing for a more lenient assessment of correctness. As shown in Eq.1, an answer is considered correct if the difference falls within this predefined threshold, which is determined by either the frames per second (FPS) or a combination of the maximum frame number and video duration.

$$\text{R} = \begin{cases} \text{True,} & \text{if } |\hat{t} - t_{\text{gt}}| \leq \tau_{ts} \\ \text{False,} & \text{otherwise} \end{cases} \text{, where } \tau_{ts} = \min\left(\frac{1}{\text{FPS}}, \frac{\text{video\_duration}}{\text{max\_frame}}\right) \tag{1}$$

where $R$ indicated the discriminant result, $\hat{t}$ indicates the time stamp extracted from the model output, $t_{\text{gt}}$ indicates the ground truth time stamp and $\tau_{ts}$ indicates the threshold.

Similar to LongVALE (Geng et al., 2024), for time span-based open-ended Grounding Q&A pairs, we evaluate correctness using the Intersection over Union (IoU) between the predicted and ground-truth time intervals, as detailed in Equation 2. For our evaluation, $\tau_{time\_span}$ was set to 0.5.

$$\text{R} = \begin{cases} \text{True,} & \text{if IoU}(\hat{I}, I_{\text{gt}}) \geq \tau_{time\_span} \\ \text{False,} & \text{otherwise} \end{cases} \tag{2}$$

where $\hat{I}$ indicates the time span extracted from the model output, $I_{\text{gt}}$ indicates the ground truth time span and $\tau_{time\_span}$ indicates the threshold.

### 4.1.2 EVALUATION FOR COUNTING AND OTHER OE Q&AS

For counting open-ended tasks, like object or action counting, LLMs are used to precisely extract numerical values from the model outputs. These extracted values are then directly compared to the ground truth: a match indicates a correct response, while any mismatch is considered incorrect.

For other open-ended tasks, we leverage LLMs to compute the similarity between the model outputs and the ground truth. Answers are then assigned a score, a floating-point number between 0 and 1, where 1 signifies a completely correct answer and 0 denotes a completely incorrect one.

### 4.2 MAIN RESULTS ON OMNIEVAL

The comprehensive evaluation results on OmniEval are presented in Table 4 and 5. Table 4 details MLLM performance across three target categories (perception, comprehension, and reasoning), whereas Table 5 highlights language-specific performance (English and Chinese). Both tables further delineate MLLM performance on open-ended (OE) and multiple-choice (MC) question formats.

As Table 4 shows, gemini-2.5-pro-preview-05-06 achieved the highest overall score of 65.20, leading particularly in OE Q&As. Qwen2.5-Omni-7B followed with an overall score of 59.57, generally outperforming its 3B counterpart (specifically, Qwen2.5-Omni-3B with 1fps achieved 54.11 overall, and Baichuan-Omni 1.5 with 64 frames achieved 50.29 overall). MiniCPM-O 2.6 scored 26.18 overall, and ViTA-1.5 scored 8.42 overall, showing comparatively lower performance. It is worth noting that ViTA-1.5 encounters tensor size out of range issues when receiving video and audio information with a sample length of over about 200 seconds simultaneously. In addition, Minicpm-o also encounters size mismatch issues on some test cases.

Gemini-2.5-pro-preview-05-06 demonstrates robust bilingual capabilities, achieving 65.02 (EN) and 65.50 (CN) overall scores. MiniCPM-O 2.6 uniquely excels in Chinese (50.87 overall, driven by 64.99 MC) compared to English (11.52). Qwen2.5-Omni models perform strongly in both languages (7B: 58.05 EN, 62.13 CN; 3B: 53.43 EN, 55.26 CN). Baichuan-Omni-1.5 shows moderate performance (51.06 EN, 48.84 CN), while ViTA-1.5 lags significantly (1.28 EN, 20.50 CN).

These results underscore the advanced capabilities of models like Gemini 2.5 Pro on complex multi-modal tasks, highlighting their superior performance and robust bilingual support on OmniEval.

### 4.3 IMPACT OF VISUAL INFORMATION AND AUDIO INFORMATION

In light of the significant performance disparities observed in the preceding evaluation, we further investigate how different types of modality-specific data contribute to the overall performance of

open-source MLLMs. Specifically, we analyze the impact of visual, auditory, and multimodal inputs on task outcomes, aiming to understand the relative importance and interplay of each modality. This exploration provides valuable insights into the data composition and modality balance required for effective multimodal understanding.

Table 6: Impact of visual information for MLLMs.

| Methods | Perception | | | Understanding | | | Reasoning | | | Overall | | |
|---|---|---|---|---|---|---|---|---|---|---|---|---|
| | Audio | +Caption | +Video | Audio | +Caption | +Video | Audio | +Caption | +Video | Audio | +Caption | +Video |
| Qwen2.5-Omni-7B (Xu et al., 2025) | 45.58 | 68.49 | 56.60 | 40.76 | 53.79 | 56.92 | 73.81 | 75.37 | 76.58 | 47.77 | 63.08 | 59.57 |
| Qwen2.5-Omni-3B (Xu et al., 2025) | 44.02 | 68.01 | 52.13 | 39.35 | 51.26 | 49.80 | 72.39 | 75.55 | 73.59 | 45.99 | 61.67 | 54.11 |
| Baichuan-Omni-1.5 (Li et al., 2025a) | 42.20 | 50.95 | 47.85 | 38.09 | 42.75 | 47.39 | 69.01 | 53.54 | 65.97 | 44.25 | 47.70 | 50.23 |
| MiniCPM-O 2.6 (OpenBMB Team, 2025) | 39.88 | 61.03 | 23.18 | 36.72 | 48.06 | 30.31 | 59.97 | 59.05 | 22.57 | 41.24 | 54.90 | 26.18 |
| VITA-1.5 (Fu et al., 2025) | 17.97 | 29.26 | 9.11 | 14.88 | 24.80 | 8.67 | 22.51 | 24.63 | 6.39 | 17.16 | 26.60 | 8.42 |

### 4.3.1 VISUAL INFORMATION.

To assess the contribution of visual information, experiments were conducted across three input modalities: audio-only, audio augmented with captions, and audio augmented with visual frames. As presented in Table 6, the incorporation of captions consistently yields a substantial enhancement in model performance across all evaluated methods. For example, Qwen2.5-Omni-7B exhibited an increase in overall score from 47.77 (audio-only) to 63.08 (audio with captions). Conversely, the subsequent addition of raw video frames generally did not lead to further improvements; in several instances, it resulted in performance degradation, which indicates weakness in aligning video with audio information. This phenomenon, notably observed with MiniCPM-O 2.6 (overall score decreasing from 54.90 to 26.18 with video addition), suggests that under the current evaluation paradigm, these MLLMs more effectively leverage textual captions than raw video content.

### 4.3.2 AUDIO INFORMATION.

To assess audio information's impact, we evaluated three input configurations: video-only, video+subtitles, and video+audio. Table 7 demonstrates that subtitles consistently enhance performance (e.g., Qwen2.5-Omni-7B's overall score increased from 47.49 to 59.57). Conversely, adding raw audio yields mixed results; some models improve (e.g., Baichuan-Omni-1.5 overall: 41.78 to 50.23), while others degrade (e.g., MiniCPM-O 2.6 overall: 38.67 to 26.18). It is worth mentioning that both MiniCPM-o and VITA-1.5 are affected by the engineering problems when merging video information and audio information for inference. This indicates that the multimodal understanding of raw audio by current MLLMs still requires significant advancement.

Table 7: Impact of audio information for MLLMs.

| Methods | Perception | | | Understanding | | | Reasoning | | | Overall | | |
|---|---|---|---|---|---|---|---|---|---|---|---|---|
| | Video | +Subtitle | +Audio | Video | +Subtitle | +Audio | Video | +Subtitle | +Audio | Video | +Subtitle | +Audio |
| Qwen2.5-Omni-7B (Xu et al., 2025) | 46.93 | 59.83 | 56.60 | 47.00 | 60.37 | 56.92 | 50.90 | 81.71 | 76.58 | 47.49 | 63.19 | 59.57 |
| Qwen2.5-Omni-3B (Xu et al., 2025) | 40.70 | 57.37 | 52.13 | 41.21 | 57.73 | 49.80 | 45.43 | 79.19 | 73.59 | 41.53 | 60.65 | 54.11 |
| Baichuan-Omni-1.5 (Li et al., 2025a) | 39.62 | 48.11 | 47.85 | 42.29 | 46.90 | 47.39 | 46.50 | 66.03 | 65.97 | 41.78 | 50.11 | 50.23 |
| MiniCPM-O 2.6 (OpenBMB Team, 2025) | 38.54 | 51.60 | 23.18 | 39.05 | 50.97 | 30.31 | 38.61 | 66.37 | 22.57 | 38.67 | 53.39 | 26.18 |
| VITA-1.5 (Fu et al., 2025) | 11.54 | 12.47 | 9.11 | 10.34 | 12.44 | 8.67 | 6.57 | 6.66 | 6.39 | 10.19 | 11.50 | 8.42 |

## 5 CONCLUSION

In this paper, we introduced OmniEval, a refined video understanding benchmark meticulously designed to address the significant limitations of current evaluation methodologies. OmniEval distinguishes itself through several key contributions: its inherent bilingual support (English and Chinese) enables the crucial direct evaluation of multilingual omni-modal models, a capability largely absent in predominantly English-centric benchmarks. Furthermore, the benchmark's substantial inclusion of open-ended questions facilitates a more comprehensive and nuanced assessment of Omni-modal Large Models' generative capabilities, moving beyond the constraints of benchmarks heavily reliant on multiple-choice formats. Finally, the explicit and granular integration of event grounding provides a targeted evaluation of these models' ability to precisely connect answers to specific video moments, thereby advancing beyond coarser temporal localization approaches. Collectively, OmniEval offers a valuable and complementary resource for the research community, fostering more nuanced and holistic progress in the challenging domain of video understanding.

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
