# OpenReview forum: "OmniEval: A Benchmark for Evaluating Omni-modal Models with Visual, Auditory, and Textual Inputs"
_ICLR.cc/2026/Conference — ICLR 2026 Conference Withdrawn Submission_

### Official Review · Reviewer_sdL2 · 2025-10-29

**Soundness:** 2
**Presentation:** 2
**Contribution:** 2
**Rating:** 2
**Confidence:** 5

**Summary:**

This paper introduces the OmniEval benchmark, designed for evaluating a model's understanding of synchronized audio-visual content. The videos are sourced from existing datasets (Finevideo and Youku-mplug). The authors' pipeline involves first using automated tools to generate visual captions and ASR transcripts for the videos. These textual annotations are then used as input for LLMs to generate question-answer pairs, which are subsequently refined by human annotators. The paper also introduces a "Grounding" task to test precise spatio-temporal understanding.

**Strengths:**

- Bilingual Benchmark: OmniEval is a bilingual video understanding benchmark that includes both English and Chinese videos and questions, which is valuable for evaluating multilingual models.
- Audio-Visual Grounding Task: OmniEval introduces the "Grounding" task, which is an important capability for audio-visual understanding.

**Weaknesses:**

- Missing Comparison with Existing Benchmarks: For the evaluation of audio-visual video understanding, there are already established benchmarks (e.g., AVUT, DailyOmni), which are not discussed or compared in the paper.
- Limitations of the Data Generation Methodology: The method of using an LLM to generate questions based on video captions and audio subtitles, while cost-effective, has several critical limitations:
  - Based on empirical evidence, Q&A pairs generated by LLMs tend to be of limited difficulty.
  - Since the visual and audio information are provided as decoupled text streams (captions and subtitles), the LLM is likely to generate questions that are superficial, touching only on the surface-level content of each modality independently. This method fails to produce questions that probe a deeper, more challenging understanding derived from the interplay between audio and video.
  - Furthermore, the synchronicity between the visual and auditory events is not considered during question generation, which undermines the benchmark's core purpose of evaluating omni-modal inputs. This oversight could even lead to questions with errors or ambiguous answers.
- Narrow Focus on ASR for Audio: Regarding the audio modality, the benchmark appears to focus almost exclusively on ASR (speech content), neglecting the role of general audio events (e.g., environmental sounds, music, sound effects), which are crucial for comprehensive scene understanding.
- Lack of Detail on Grounding Annotation: The annotation process for the Grounding task is not described. The paper provides few details on how these temporal annotations were created or how their accuracy and consistency were ensured.
- Incomplete Experiments: The experimental section is insufficient. The paper only compares several omni-LLMs but lacks a crucial baseline of strong vision-only LLMs with subtitles as an additional text input. Moreover, the paper fails to compare against other powerful audio-visual LLMs such as Video-LLaMA 2 and Video-SALMONN 2.

**Questions:**

- What is the performance of strong vision-only LLMs (provided with subtitles as an additional input) on the OmniEval benchmark?
- Which LLM is used for annotation in Section 3?

---

### Official Review · Reviewer_hcPT · 2025-10-30

**Soundness:** 2
**Presentation:** 2
**Contribution:** 2
**Rating:** 2
**Confidence:** 4

**Summary:**

This paper introduces OmniEval, a comprehensive benchmark designed to evaluate omni-modal large language models (MLLMs), especially those capable of processing text, images, and audio. The benchmark consists of diverse tasks covering perception, understanding, and reasoning across modalities.

**Strengths:**

1. The proposed benchmark spans text, video, and audio, making it  reflective of real-world multimodal task needs.

2. The proposed benchmark includes both English and Chinese videos, posing additional challenges for omni-foundation models.

**Weaknesses:**

1. A detailed comparison of different MLLMs on 12 tasks is missing.

2. Qualitative comparison of different MLLMs on the proposed benchmark is missing.

3. The authors highlighted temporal grounding as a key feature of this benchmark, but I could not find how MLLMs perform on this task in the experiments.

**Questions:**

1. L214-215, what does it mean by generating the captions using appropriate methods? what appropriate methods?

2. What is the LLM used for evaluating open-ended QAs?

3. Table 7 shows that audio is useful in current MLLMs. Can you give some QA examples where audio information is important in answering the question, especially in the Grounding task.

---

### Official Review · Reviewer_siC2 · 2025-10-30

**Soundness:** 3
**Presentation:** 2
**Contribution:** 3
**Rating:** 6
**Confidence:** 4

**Summary:**

The paper proposes a bilingual (EN/CN) benchmark: OmniEval, for omni-modal models that process video + audio + text jointly. The suite contains 780 synchronized videos and 2,411 QA items spanning open-ended (1,278) and multiple-choice (1,133) formats. These are categorized into 3 major task families and 12 sub-tasks, with a special temporal grounding task (moment and time-span) intended to probe fine-grained spatiotemporal understanding. The construction pipeline aggregates videos (YouTube/Youku; partly via FineVideo and Youku-mplug), obtains captions/subtitles and ASR transcripts, filters out low-speech videos, then generates and manually curates QAs. The evaluation reports baselines for Qwen2.5-Omni, Baichuan-Omni, MiniCPM-O, VITA-1.5, and Gemini 2.5 Pro, including language-wise performance and ablations isolating the contribution of captions, audio, and raw video; captions typically help the most, while adding raw video sometimes degrades scores under the current pipelines. Grounding OE grading uses time-tolerance for moments and IoU≥0.5 for spans. Overall, Gemini 2.5 Pro tops the table; Qwen2.5-Omni-7B is the strongest open model.

**Strengths:**

1) Unlike vision-only or audio-text setups, OmniEval evaluates joint A+V+T reasoning, with both English and Chinese coverage, which is still underexplored.

2) he mix of OE (1,278) and MC (1,133), distributed across 12 sub-tasks, supports both generative analysis and standardized accuracy comparisons; the Grounding category (moment/time-span) is a good addition.

3) The adaptive timestamp tolerance and IoU≥0.5 criteria for OE grounding are explicit and easy to re-implement, which helps reproducibility of temporal evaluation.

4) Tables 6–7 show that captions/subtitles consistently lift performance, whereas adding raw video (or audio) can be mixed or negative. This seems to provide a useful empirical signal for the community about current model bottlenecks

5) The paper has good coverage of experiments. They evaluate multiple open source and proprietary models.

**Weaknesses:**

1) The pipeline excludes low-speech videos (ASR subdensity < 0.5), which systematically under-samples silent, music-dominant, or non-verbal soundscapes. This may bias the benchmark towards text-anchored items and may under-stress purely audio-visual fusion. A short analysis of discarded vs kept videos (content type, duration, genre) would help to clarify the bias.

2) OE scoring uses an LLM-as-judge/extractor but the paper provides no agreement statistics (e.g., κ with CIs) or dual-judge disagreement audit.

3) The time tolerance τ scales with FPS or (duration / max frames). While practical, this makes correctness thresholds dataset- and setting-dependent; a brief sensitivity table (vary τ, IoU) would calibrate how robust rankings are to scoring hyperparameters.

4) The ablations indicate captions lift scores far more reliably than adding frames; in several models, adding video reduces accuracy. This suggests tasks often remain text-solvable, weakening claims of deep A+V synergy. Perhaps, including a "video-only” diagnostic track (and per-subtask breakdown) would help to separate textual vs visual competence.

**Questions:**

1) Could you report κ with 95% CIs for OE scoring, stratified by Perception/Understanding/Reasoning and by language (EN/CN)? A small stratified audit would bring more confidence.

2) Would you consider reportingASR WER (or a proxy) for EN and CN, and show sensitivity of results to ASR errors on speech-heavy tasks?

3) Could you please add per-source (YouTube/Youku/FineVideo/Youku-mplug) and per-genre accuracy tables to reveal  any inheritance effects?

4) Could you include a short analysis varying τ (moment) and IoU threshold (span) to show ranking stability?

5) Since OmniEval includes tasks with differing temporal scales (moment-level vs. span-level grounding), can the authors quantify temporal granularity sensitivity, e.g., whether models perform uniformly across events lasting <3s versus >15s?

6) Given that the bilingual corpus sources (YouTube/Youku) may differ culturally, do accuracy disparities correlate with genre or region?

---

### Official Review · Reviewer_3gEp · 2025-10-31

**Soundness:** 3
**Presentation:** 3
**Contribution:** 3
**Rating:** 6
**Confidence:** 4

**Summary:**

This paper introduces OmniEval, a new benchmark designed to evaluate the ability of omni-modal models to simultaneously understand visual, auditory, and textual inputs. The main contributions are threefold:
1. Addresses an Evaluation Gap: OmniEval focuses on "full-modal collaboration," with tasks meticulously designed to require a strong coupling of information from multiple modalities (video + audio + text) to answer correctly. This addresses the limitation of existing benchmarks that often evaluate modalities in isolation.
2. High-Quality Data Curation: OmniEval consists of 2,617 Q&A pairs (in both Chinese and English). Its curation pipeline (Section 3.3) combines "automated filtering" (removing overly simple samples) with critical "manual judgment" (ensuring multi-modal dependency), ensuring the benchmark's difficulty and validity.
3. Reveals Deep Model Flaws: Using this benchmark, the paper's modal ablation study (Section 4.3) discovers that current SOTA omni-modal models struggle significantly with fusing raw audio-visual signals. They exhibit an over-reliance on "text" (subtitles/captions) , and their performance can even degrade significantly when forced to process raw video frames or audio.

**Strengths:**

1. This paper accurately identifies the "blind spot" in current omni-modal evaluation—namely, the lack of assessment for synergistic understanding. The paper's core, original concept is its "full-modal collaboration" evaluation philosophy.
2. The inclusion of bilingual (CN/EN) support and the fine-grained "Grounding" (temporal localization) task  effectively fills gaps left by existing benchmarks.
3. The hybrid pipeline described in Section 3.3 is excellent. The "Judgment" step (to ensure multi-modal dependency) and the "Distribution" correction step (to fix LLM-generation biases)  are particularly crucial for ensuring the benchmark's quality and validity, setting it far apart from simple "collect-and-label" efforts.

**Weaknesses:**

1. The benchmark contains a total of 2,617 Q&A pairs derived from 810 videos. When these are divided among 3 major categories, 12 sub-task types, and 2 languages, the number of samples for each fine-grained task (e.g., "Grounding" in Chinese) may be very small. This raises concerns about the statistical significance of the evaluation results. For example, the "Grounding" task has only 342 pairs in total; further subdivision by language and format (OE/MC) may result in an insufficient sample size for robust conclusions.
2. The paper's core claim of "full-modal collaboration" relies on a manual "Judgment" step to determine how many modalities are required to answer a question. This is an inherently subjective task. The paper lacks a detailed protocol for this step. For instance, what was the Inter-Annotator Agreement (IAA)? How were disagreements (e.g., whether audio was truly necessary) resolved? Without IAA data, the reliability of this core claim is questionable.

**Questions:**

1. As mentioned in Weakness #1, the 2,617 Q&A pairs might be spread too thin across 12 sub-tasks and 2 languages Could the authors please provide a detailed table showing the Q&A distribution across (12 sub-tasks $\times$ 2 languages $\times$ 2 formats)? For categories with a small sample size (e.g., "Grounding" 28), how do the authors ensure the statistical robustness of the evaluation results?
2. As mentioned in Weakness #2, the manual "Judgment" step is central to the paper's methodology. Could the authors please provide details on the annotation guidelines for this task? Was Inter-Annotator Agreement (IAA) calculated for this step? If so, what was the score (e.g., Cohen's Kappa or Krippendorff's Alpha), and how were disagreements among annotators resolved?

---

### Note · Authors · 2025-11-12

I have read and agree with the venue's withdrawal policy on behalf of myself and my co-authors.